# Increase in IGF-1 Expression in the Injured Infraorbital Nerve and Possible Implications for Orofacial Neuropathic Pain

**DOI:** 10.3390/ijms20246360

**Published:** 2019-12-17

**Authors:** Shiori Sugawara, Masamichi Shinoda, Yoshinori Hayashi, Hiroto Saito, Sayaka Asano, Asako Kubo, Ikuko Shibuta, Akihiko Furukawa, Akira Toyofuku, Koichi Iwata

**Affiliations:** 1Department of Psychosomatic Dentistry, Graduate School of Medical and Dental Science, Tokyo Medical and Dental University, Tokyo 113-8510, Japan; 0629ompm@tmd.ac.jp (S.S.); toyoompm@tmd.ac.jp (A.T.); 2Department of Physiology, Nihon University School of Dentistry, Tokyo 101-8310, Japan; hayashi.yoshinori@nihon-u.ac.jp (Y.H.); kubo.asako@nihon-u.ac.jp (A.K.); shibuta.ikuko@nihon-u.ac.jp (I.S.); iwata.kouichi@nihon-u.ac.jp (K.I.); 3Department of Complete Denture Prosthodontics, Nihon University School of Dentistry, Tokyo 101-8310, Japan; saitou.hiroto85@nihon-u.ac.jp; 4Department of Oral Diagnostic Sciences, Nihon University School of Dentistry, Tokyo 101-8310, Japan; bunny2838@gmail.com; 5Department of Clinical Medicine, Nihon University School of Dentistry, Tokyo 101-8310, Japan; furukawa.akihiko@nihon-u.ac.jp

**Keywords:** trigeminal nerve injury, transient receptor potential channels vanilloid subfamily type 2, transient receptor potential channels vanilloid subfamily type 4, mechanical allodynia, trigeminal ganglion, macrophage, upper cervical spinal cord

## Abstract

Insulin-like growth factor-1 (IGF-1) is upregulated in the injured peripheral nerve bundle and controls nociceptive neuronal excitability associated with peripheral nerve injury. Here, we examined the involvement of IGF-1 signaling in orofacial neuropathic pain following infraorbital nerve injury (IONI) in rats. IONI promoted macrophage accumulation in the injured ION, as well as in the ipsilateral trigeminal ganglion (TG), and induced mechanical allodynia of the whisker pad skin together with the enhancement of neuronal activities in the subnucleus caudalis of the spinal trigeminal nucleus and in the upper cervical spinal cord. The levels of IGF-1 released by infiltrating macrophages into the injured ION and the TG were significantly increased. The IONI-induced the number of transient receptor potential vanilloid (TRPV) subfamily type 4 (TRPV4) upregulation in TRPV subfamily type 2 (TRPV2)-positive small-sized, and medium-sized TG neurons were inhibited by peripheral TRPV2 antagonism. Furthermore, the IONI-induced mechanical allodynia was suppressed by TRPV4 antagonism in the whisker pad skin. These results suggest that IGF-1 released by macrophages accumulating in the injured ION binds to TRPV2, which increases TRPV4 expression in TG neurons innervating the whisker pad skin, ultimately resulting in mechanical allodynia of the whisker pad skin.

## 1. Introduction

Trigeminal nerve injury due to maxillary bone fracture, tooth extraction, or dental implant placement is considered one of the causes of neuropathic pain in the orofacial region [1,2]. Patients with orofacial neuropathic pain are affected by prolonged intractable mechanical or thermal hypersensitivities in the orofacial region after injury, and clinicians struggle to alleviate these pain hypersensitivities because the detailed mechanisms underlying this neuropathic pain have not been fully elucidated yet [3].

It has been reported that peripheral nerve injury induces in the region of the nerve injury infiltration and proliferation of macrophages, which are involved in the regeneration of the injured peripheral nerve [4,5]. Furthermore, pain hypersensitivity following peripheral nerve injury is abolished after deletion of the macrophages accumulated in the nerve-injured region or the sensory ganglion [6,7]. It is also known that insulin-like growth factor-1 (IGF-1), a polypeptide growth factor highly similar in sequence to insulin, plays an essential role in the maintenances of various neuronal functions in the central and peripheral nervous system [8,9]. Besides, IGF-1 receptor (IGF-1R) activation by IGF-1 released from infiltrating macrophages induces translocation of transient receptor potential (TRP) vanilloid subfamily type 2 (TRPV2) channel to the cell membrane, resulting in noxious heat or mechanical hypersensitivity [10,11].

TRPV2 as a member of the TRP channel family is a calcium-permeable channel and mostly expressed in medium- to large-sized sensory neurons with Aδ fibers sensitive to mechanical and heat (above 52 °C) stimuli [12,13]. In our previous study, increased TRPV2 expression in trigeminal ganglion (TG) neurons innervating the orofacial traumatized regions caused trauma-induced mechanical hypersensitivity [14]. Moreover, calcium influx into nociceptive neurons leads to the activation of neuronal calcium/calmodulin-dependent kinase II (CaMKII), resulting in neuropathic pain due to hyperexcitability of nociceptive neurons [15]. These results imply that IGF-1 released by accumulated macrophages in the region of a nerve injury or in the TG regulates the excitability of injured trigeminal neurons, playing a pivotal role in neuropathic pain conditions.

The aim of the present study was first to evaluate the contribution of IGF-1 to orofacial neuropathic pain after infraorbital nerve injury (IONI). We examined the effects of TRPV2 antagonism on orofacial mechanical hypersensitivity following IONI and quantified IGF-1 expression in the injured ION and TG. Finally, we assessed the effects of TRPV2 antagonism on the expression of the TRP vanilloid subfamily type 4 (TRPV4) channel which is involved in nociceptive mechanical sensitivity of TG neurons innervating the orofacial region.

## 2. Results

### 2.1. Effect of TRPV2 Antagonism on Mechanical Allodynia and the Activity of Wide Dynamic Range (WDR) Neurons in the Upper Cervical Spinal Cord (Vc/C1-C2) Following IONI

The mechanical head withdrawal threshold (MHWT) was significantly decreased on day 1 after IONI, and this MHWT decrease persisted until day 7. By contrast, there were no changes in the MHWT after sham treatment over the entire experimental period. The decrease in MHWT between day 1 and day 5 after IONI was partially reversed by tranilast administration to the whisker pad skin (Figure 1).

A total of 33 nociceptive neurons were recorded in the Vc/C1-C2 and classified as WDR neurons according to their electrophysiological properties in response to mechanical stimulation of the whisker pad skin (sham: 11 neurons, IONI with vehicle: 11 neurons, IONI with tranilast: 9 neurons). Figure 2A demonstrates typical von Frey filament-, brush-, and pinch-evoked responses of Vc/C1-C2 WDR neurons on day 3 after IONI with tranilast or vehicle administration or after sham treatment. All WDR neurons in the IONI with tranilast, IONI with vehicle, and sham treatment groups increased their firing frequencies following an increase in mechanical stimulus intensities by the von Frey filaments. Tranilast administration completely reversed the IONI-induced increase in background-corrected firing rate of WDR neurons in response to sequential mechanical stimulation with von Frey filaments (Figure 2B). Furthermore, the neuronal background activity, as well as the brush- and pinch-evoked responses, of Vc/C1-C2 WDR neurons were significantly increased on day 3 after IONI (sham: 1.68 ± 1.37, IONI with vehicle administration: 22.33 ± 1.90, IONI with Tranilast, administration: 0.46 ± 0.14), and these increases in neuronal firing frequencies were entirely prevented by tranilast administration (Figure 2C–E).

### 2.2. IGF-1 Expression in the Injured ION and TG

In the injured ION, IGF-1 was only expressed in Iba1-immunoreactive (IR) cells. By contrast, IGF-1 was not expressed in glial fibrillary acidic protein (GFAP)-IR cells on day 3 and day 7 following IONI or sham treatment (Figure 3A). In western blots, the relative amount of IGF-1 protein was also increased in the injured ION on day 3 following IONI (sham: 0.042 ± 0.015, IONI: 0.17 ± 0.03) (Figure 3B). Similarly, in the TG ipsilateral to the injured ION, IGF-1 was expressed in Iba1-IR cells but not in GFAP-IR cells, on day 3 following IONI and sham treatment (Figure 3C). Furthermore, the relative amount of IGF-1 protein was also increased in the ipsilateral TG following IONI in comparison to sham treatment (sham: 0.18 ± 0.03, IONI: 0.3 ± 0.03) (Figure 3D). The relative amount of IGF-1 protein in the injured ION and TG ipsilateral to IONI recovered to the same level as sham treatment on day 7 following IONI.

### 2.3. Effects of Tranilast on the Changes in the Number of TRPV4-IR TG Neurons Expressing TRPV2 Following IONI

On day 3 after IONI with or without tranilast administration or after sham treatment, TRPV2-IR and TRPV4-IR TG neurons innervating the whisker pad skin were observed in the TG ipsilateral to the IONI or sham treatment (Figure 4A). IONI caused a significant increase in the number of FluoroGold (FG)-labeled TRPV2-IR TG neurons, whereas the number of FG-labeled TG neurons did not change. Moreover, this increase was significantly reversed by tranilast administration to the whisker pad skin (Sham: 15.71 ± 0.02%, Vehicle: 30.18 ± 0.03%, Tranilast: 24.85 ± 0.02%) (Figure 4B). The mean percentage of TRPV2/TRPV4 double-stained neurons among TRPV2-IR TG neurons in TRPV2-IR TG neurons innervating the whisker pad skin was on day 3 after IONI significantly increased relative to the sham treatment group, and this increase was also prevented by tranilast administration (Sham: 48.42 ± 0.06%, Vehicle: 78.07 ± 0.02%, Tranilast: 42.99 ± 0.04%) (Figure 4C). The cell diameter analysis indicated that following IONI, the increase in the number of TRPV4-IR and TRPV2-IR TG neurons innervating the whisker pad skin was reversed by tranilast administration mainly in the group with cell sizes of 300–699 mm^2^ (Vehicle: 20.71 ± 3.52, Tranilast: 9.52 ± 1.48) (Figure 4D).

### 2.4. Effects of TRPV4 Antagonism on the Decreased MHWT Following IONI

To assess the effect of TRPV4 antagonism on the decreased MHWT following IONI, changes in the MHWT on day 3 after IONI were examined following HC-067047 administration to the whisker pad skin. The decreased MHWTs were significantly recovered 30 min after HC-067047 administration, but this recovery did not last until 60 min after HC-067047 administration (Vehicle: 16.32 ± 5.70 g, HC-067047: 39.16 ± 3.38 g) (Figure 4E). 

### 2.5. Effects of IGF-1 Neutralization on the Decreased MHWT Following IONI

To assess the effect of IGF-1 on the decreased MHWT following IONI, changes in the MHWTs after IONI by IGF-1 neutralization were examined. The decreased MHWTs were significantly recovered by daily IGF-1 neutralizing antibody administration (Day 1, sham-normal IgG: 76.00 ± 8.76 g, ION-normal IgG: 11.40 ± 3.36 g, IONI-IGF-1 neutralizing antibody: 45.0 ± 5.48 g) (Figure 5).

## 3. Discussion

Following peripheral nerve injury, various immune cells such as macrophages, monocytes, lymphocytes, or neutrophils infiltrate the inflamed tissue and the corresponding sensory ganglion. These immune cells play an essential role in the promotion of tissue regeneration [7,16,17]. The C-C chemokine 2 (CCL2) known as monocyte chemotactic protein-1 can be produced by various cells such as fibroblasts, epithelial cells, basophils, or monocytes [18,19,20]. In various disease models, the inflammatory conditions are known to induce focal infiltration of monocytes which produce CCL2, and it is reported that the CCL2 levels in the inflamed tissue consequently increase [21,22]. Following peripheral nerve injury, injured neurons and Schwann cells also release CCL2 [23], and CCL2 signaling induces activation and increase in the number of macrophages in sensory ganglia [24,25]. The C-C chemokine receptor 2 (CCR2) to which CCL2 binds preferentially is expressed in infiltrating macrophages that express genes typical for classical macrophage activation [26]. Moreover, infiltrating macrophages release IGF-1 via CCL2-CCR2 signaling in the inflamed tissue following tissue injury and in sensory ganglia following peripheral nerve injury [27]. In response to injury of muscle tissue, macrophages infiltrating the injured site have also been reported to release IGF-1 [28]. In the present study, IONI promoted macrophage infiltration into the injured ION and the corresponding TG. The amount of IGF-1 released by macrophages infiltrating the ION and TG was significantly increased on day 3 following IONI. Furthermore, IGF-1 neutralization partially depressed IONI-induced mechanical hypersensitivity. Taken together, our findings suggest that the infiltrating macrophages activated via CCL2-CCR2 signaling release IGF-1 following IONI, resulting in increased IGF-1 levels in the injured ION site and the ipsilateral TG. Increased IGF-1 may play an important role in IONI-induced mechanical hypersensitivity. However, it is expected that hundreds of different cytokines and growth factors which are released in the injured ION and the corresponding TG, further analysis is necessary. 

Similar to TRPV2, TRPV4 is also a member of the vanilloid subfamily of TRP channels and expressed in sensory neurons [29]. Specifically, it is known that TRPV4 is essential for the detection of innocuous and noxious mechanical stimuli [30,31]. TRPV4 is involved in the enhanced sensitivity of C-fibers to noxious mechanical stimuli applied to inflamed tissue [32]. TRPV2 is also referred to as a growth factor-regulated channel in addition to be a nociceptor for noxious heat and mechanical stimuli since IGF-1 signaling upregulates the expression of the TRPV2 on the membrane in cultured cells [33]. The TRPV2-mediated increases in intracellular Ca^2+^ lead to enhanced tumor necrosis factor (TNF)-α expression via nuclear factor-kappa B signaling [34]. Moreover, TNF-α signaling reportedly triggers increases in TRPV4 mRNA and protein expression in cultured cells [35]. In the current study, IONI caused a significant increase in the number of TRPV2-positive TG neurons innervating the whisker pad skin, and this increase was prevented by peripheral TRPV2 antagonism. Additionally, about half of the TRPV2-positive TG neurons innervating the whisker pad skin expressed also TRPV4, and the TRPV4 expression in TRPV2-positive TG neurons innervating the whisker pad skin was significantly increased after IONI. The IONI-induced TRPV4 upregulation in TRPV2-positive small- and medium-sized TG neurons was also reversed by peripheral TRPV2 antagonism. Furthermore, the IONI-induced mechanical allodynia was inhibited by TRPV4 antagonism in the whisker pad skin. Taken together, these results suggest that IGF-1 signaling via TRPV2 in TG neurons innervating the whisker pad skin increases intracellular Ca^2+^ concentrations leading to enhanced TNF-α expression after IONI. The elevated TNF-α signaling causes an increase in TRPV4 expression in TRPV2-positive TG neurons, resulting in IONI-induced mechanical allodynia of the whisker pad skin and neuronal hyperexcitability of the TG due to the upregulated TRPV4 expression. Although mechanical allodynia was induced in the whisker pad skin, the IONI-induced increases in activity of WDR neurons in the Vc/C1-C2 both spontaneous and in response to innocuous and noxious mechanical stimulation were significantly suppressed by TRPV2 antagonism. It is also indicative that the IONI-induced hyperexcitability of TG neurons responsible for the mechanical allodynia of the whisker pad skin depends on peripheral TRPV2 signaling.

Besides, IGF-1 does not only bind to TRPV2 channels but also to IGF-I receptors which are predominantly expressed in nociceptive neurons of sensory ganglia [36]. It has been reported that IGF-1 signaling via IGF-I receptors promotes the translocation of TRPV2 from intracellular pools towards the plasma membrane via phosphoinositide 3 (PI3) kinase activation in vitro; IGF-1-evoked Ca^2+^ entry is consequently caused by TRPV2 activation [12,33,37]. Moreover, upregulation of the PI3 kinase pathway promotes TRPV2 activity [38]. The translocation and hyperactivity of TRPV2 channels via the activity of the PI3 kinase cascade may also enhance Ca^2+^ entry leading to increased TNF-α expression, thus promoting the increase in TRPV4 expression in TRPV2-positive TG neurons innervating the whisker pad skin. Furthermore, TRPV4 activation on the cell surface leads to PI3 kinase signaling [39], and the Ca^2+^-permeable TRPV4 channel is activated by upregulation of the PI3 kinase pathway [40]. Although further studies are required, the observed IONI-induced mechanical allodynia may be caused not only by an increase in TRPV4 expression in TG neurons innervating the whisker pad skin but also by TRPV4 activation via upregulation of the PI3 kinase pathway.

In conclusion, these results suggest that IGF-1 released by macrophages in the injured ION and the corresponding TG anticipates TRPV4 activation in TG neurons innervating the whisker pad skin via IONI-induced TRPV2 signaling, which is a potential mechanism of mechanical allodynia of the whisker pad skin. Thus, changes in the characteristics of TRPV4 channels via IGF-1 signaling may be a promising therapeutic target for the treatment of orofacial neuropathic pain.

## 4. Materials and Methods

### 4.1. Animals

One hundred twenty-six male Sprague-Dawley rats (Japan SLC, Hamamatsu, Japan) weighing 150-200 g were used in this study. The animals were housed with food and water ad libitum, and animal breeding cages were remained in controlled climate and light (23 °C, 12-h:12-h light-dark cycle, light on at 7:00). The animal protocols in this study were approved by the animal experimentation committee of Nihon University on 19 October 2018 (AP18DEN025-1), and experimental procedures were performed according to the National Institutes of Health Guide for the Care and the Use of Laboratory Animals (PHS Law 99-158, revised 2002) and the guidelines of the International Association for the Study of Pain [41].

### 4.2. Infraorbital Nerve Injury

Initially, the animals were kept on a warm mat (37 °C) and profoundly anesthetized with an intraperitoneal (i.p.) injection of a mixture composed of butorphanol (2.5 mg/kg; Meiji Seika Pharma, Tokyo, Japan), midazolam (2.0 mg/kg; Sandoz, Tokyo, Japan), and medetomidine (0.15 mg/kg; Zenoaq, Fukushima, Japan). The infraorbital nerve injury (IONI) procedure was conducted as formerly reported [42]. A small incision (10 mm) of the right buccal mucosa along the gingiva-buccal margin proximal to the first molar was made with a scalpel, and the right ION bundle was exposed by removal of the surrounding connective tissue. One-third of the ION bundle was ligated tightly with a 6-0 silk thread (Natsume, Tokyo, Japan), and the incision was closed with another 6-0 silk thread. For control, the sham procedure was carried out which was completely identical except for the ligation of the ION bundle. In all experiments, numbers and distress of the animals used were minimized.

### 4.3. Mechanical Sensitivity Measurement of the Whisker Pad Skin

The mechanical head withdrawal threshold (MHWT) of the whisker pad skin was measured as previously described [43]. Briefly, unrestrained animals were initially trained to stick their snouts through a small hole in the cage for mechanical stimulations of the whisker pad skin using von Frey filaments (Touch-Test Sensory Evaluator; North Coast Medical, Morgan Hill, CA) on seven consecutive days. Mechanical stimulation was applied to the whisker pad skin using the von Frey filaments in ascending order of mechanical intensity (1, 4, 6, 15, 26, 35, 60, and 100 g) five times at one-minute intervals, and the MHWT for each animal was determined as the lowest mechanical intensity that elicited a head withdrawal response three or more times out of five stimuli. After MHWT measurements of the whisker pad skin provided stabile results, IONI or sham treatment was conducted as described above. Afterward, MHWT measurements were repeated under the same conditions before and every other day for 15 days after surgery. The MHWT measurements were performed under blinded conditions. 

### 4.4. Drug Administration

To assess the involvement of TRPV2 signaling in IONI-induced mechanical hypersensitivity of the whisker pad skin, 25 µL of either the TRPV2 antagonist 2-[[3-(3,4-dimethoxyphenyl)-1-oxo-2-propen-1-yl]-benzoic acid (1.2 µg/µL, tranilast; Cayman, MI, USA) dissolved in 20% dimethyl-sulfoxide (DMSO) in saline or vehicle (20% DMSO in saline) was subcutaneously injected into the whisker pad skin. The drugs were administered under inhalation anesthesia using 1.5% isoflurane (Mylan, Canonsburg, PA) once a day until day 15 following IONI using a 27-gauge needle. The MHWT was measured before and every other day for 15 days following IONI or sham treatment under the same conditions as described above.

To evaluate the involvement of TRPV4 signaling in IONI-induced changes in mechanical sensitivity of the whisker pad skin, 5 µL of either the TRPV4 antagonist 2-Methyl-1-[3-(4-morpholinyl)-5-phenyl-N-[3(trifluoromethyl)phenyl]-1H-pyllore-3-carboxamide (100 µg/kg, HC067047; Abcam, Cambridge, UK) dissolved in 20% DMSO or vehicle (20% DMSO) was subcutaneously administered to the ligated ION bundle site under inhalation anesthesia using 1.5% isoflurane. The MHWTs were measured up to 60 min before HC067047 administration and every 30 min after its administration on day 3 following IONI as described above. The administered tranilast and HC067047 doses were based on previous reports [14,44].

Furthermore, to evaluate that IGF-1 signaling in IONI-induced mechanical hypersensitivity of whisker pad skin, 10µL of rabbit IGF-1 neutralizing antibody (2.0 µg/mL, ab9572; Abcam, Cambridge, UK) or normal rabbit IgG (2.0 µg/mL, #148-09551; Wako, Tokyo, Japan) dissolved in 0.01 M PBS was subcutaneously administrated into whisker pad skin under inhalation anesthesia using 1.5% isoflurane once a day until day 7 following IONI using a 27-gauge needle. The MHWT was measured before and daily for 7 days following IONI or sham treatment under the same conditions as described above. The administered rabbit IGF-1 neutralizing antibody dose was based on previous reports [45,46].

### 4.5. Immunohistochemistry

To identify trigeminal ganglion (TG) neurons that innervate the whisker pad skin, 10 µL of 4% hydroxystilbamidine (FluoroGold [FG]; Fluorochrome, Denver, CO, USA) dissolved in 4% saline was subcutaneously injected into the right whisker pad skin prior to the surgery. On day 3 following IONI or sham treatment, the rats were transcardially perfused with 4% paraformaldehyde (PFA) solution dissolved in 0.1 M phosphate buffer (pH 7.4) under i.p. deep anesthesia with the above-described solution. The right TGs were dissected and post-fixed in the 4% PFA solution for some days at 4 °C. The tissue samples were then transferred to 20% sucrose (weight percent) dissolved in 0.01 M phosphate-buffered saline (PBS) overnight for cryoprotection. On day 3 following IONI or sham treatment, the rats without FG injection were also perfused, and the right ION was dissected and post-fixed as described above. The TG and ION tissues were embedded in Tissue-Tek O.C.T. Compound (Sakura Finetek, Tokyo, Japan), cut at 10 µm thickness, mounted on Matsunami Adhesive Silane-Coated Superfrost Plus microscope slides (Matsunami, Osaka, Japan), and dried overnight at room temperature. For TG sections, 8 sections (150 µm interval) per TG were selected for analysis. Following rinsing with 0.01 M PBS, the TG and ION sections were incubated for 30 min in 50% ethanol dissolved in distilled water for antigen activation. Following rinsing with 0.01 M PBS, the TG sections were reacted with rabbit anti-TRPV2 polyclonal antiserum (1:500; Cat# AG1350; Abgent, San Diego, CA, USA) and sheep anti-TRPV4 polyclonal antiserum (1:500; Cat# OST00012W; Life Technologies, Carlsbad, CA) diluted in 0.01 M PBS with 1% skimmed milk and 0.1% Triton-X (Merck, Darmstadt, Germany) at 4 °C for 3 days. Furthermore, the ION sections were reacted with mouse anti-IGF-1 monoclonal antibody (1:1,000; ab176523; Abcam, Cambridge, UK) and rabbit anti-Iba1 polyclonal antibody (1:1,000; # 019-19741; Wako, Tokyo, Japan) or rabbit anti-glial fibrillary acidic protein (GFAP) polyclonal antibody (1:1,000; # Z0334; Agilent, Santa Clara, CA, USA) diluted in 0.01 M PBS with 1% skimmed milk and 0.1% Triton-X (Merck) at 4 °C for 3 days. Following rinsing with 0.01 M PBS, the TG and ION sections were subsequently reacted with goat anti-rabbit Alexa Fluor 568 IgG (1:200; # 1871167; Thermo Fisher Scientific, Fremont, CA, USA) and donkey anti-sheep Alexa Fluor 488 IgG (1:200; # ab150177; Abcam) or goat anti-mouse Alexa Fluor 488 IgG (1:200; # 1874804; Thermo Fisher Scientific) diluted in 0.01 M PBS with 0.1% Triton-X for 3 h at room temperature. Then, the TG and ION sections were coverslipped in mounting medium (PermaFluor, Thermo Fisher Scientific). FG-labeled TRPV2-immunoreactive (IR) and FG-labeled TRPV4-IR cells in TG sections, as well as IGF-1-Iba1 and IGF-1-GFAP co-stained cell in ION sections, were identified and analyzed using a BZ-9000 system (Keyence, Osaka, Japan) equipped with the appropriate filters. Cells with a fluorescence intensity two times or more than that of the average background signal were defined as immunoreactive. No specific immunoreactivity was detected in the absence of primary antibodies. The number of FG-labeled TRPV2-IR TG neurons and FG-labeled TRPV2-IR and TRPV4-IR neurons were calculated using the following formulas: (1) 100 × total number of FG-labeled TRPV2-IR cells / total number of FG-labeled cells and (2) (FG-labeled TRPV2-IR + FG-labeled TRPV4-IR cells) / FG-labeled TRPV2-IR cells; (all parameters determined in 4 TG sections). Regarding the FG-labeled TRPV2-IR and TRPV4-IR cells, the number of each group of cells classified by IR area (< 300 µm^2^, 300–699 µm^2^, > 700 µm^2^) was calculated according to previous reports [42,47].

### 4.6. Western Blotting

On days 3 and 7 after IONI or sham treatment, the rats were perfused with physiological saline under deep i.p. anesthesia with the above-described solution. The tissue including the injured or sham-treated ION and TG was removed immediately and homogenized in ice-cold lysis buffer (137 mM NaCl; 20 mM Tris-HCl, pH 8.0; 1% NP40; 10% glycerol; 1 mM phenylmethylsulfonyl fluoride; 10 μg/mL aprotinin; 1 g/mL leupeptin; and 0.05 mM sodium vanadate). The homogenate was centrifuged, and the supernatant was collected. The protein concentration of the supernatant was determined using a protein assay kit (Bio-Rad, Hercules, CA, USA). Supernatants were heat-denatured in Laemmli sample buffer solution (Bio-Rad), and the samples with protein adjusted to 30 μg were subjected to electrophoresis on 10% sodium dodecyl sulfate-polyacrylamide gel electrophoresis for protein separation. The samples were transferred to a polyvinylidene difluoride membrane (Trans-Blot Turbo Transfer Pack; Bio-Rad). The membrane was rinsed with Tris-buffered saline mixed with 0.1% Tween 20 (TBST) and incubated in 3% bovine serum albumin (BSA; Bovogen, Essendon, Australia). The membrane was then incubated with mouse anti-IGF-1 monoclonal antibody (1:1,000; ab176523; Abcam, Cambridge, UK) diluted in TBST with 3% BSA overnight at 4 °C. Then, horseradish peroxidase-conjugated rabbit anti-mouse antibody (Jackson Immuno Research, West Grove, PA, USA) was incubated for 2 h at room temperature. Protein-bound antibodies were detected using Western Lightning ELC Pro (PerkinElmer, Waltham, MA, USA) and visualized using a ChemiDoc MP system (Bio-Rad). Using β-actin antibody (1:200; Sc-69879, Santa Cruz, CA, USA) following removal of bound protein by a stripping reagent (Thermo Scientific), protein levels were normalized to β-actin. 

### 4.7. Neuronal Recordings in the Spinal Trigeminal Nucleus and Upper Spinal Cord

On day 3 after IONI with tranilast or vehicle administration or after sham treatment, neuronal recordings were performed in the subnucleus caudalis of the spinal trigeminal nucleus and in the upper cervical spinal cord (Vc/C1-C2) using extracellular recording procedures as reported previously [48,49]. Initially, the trachea was intubated for artificial respiration. The left jugular vein for gallamine triethiodide pancuronium bromide (1.25 mg/kg/h; Sigma-Aldrich) infusion, as well as the femoral artery for blood pressure monitoring, were cannulated under deep anesthesia induced by i.p. pentobarbital sodium (50 mg/kg). The rat was immobilized and artificially ventilated, its head was rigidly mounted in a stereotaxic apparatus, and the posterior margin of the occipital bone was removed to expose the brainstem. After removal of the dura mater, the brainstem was soaked in a pool of mineral oil contained by the surrounding skin flaps. After the preparation, the expiratory end-tidal CO_2_ (3.5–5.5%), heart rate (less than 300 times/min), and body temperature (36–38 °C) were monitored and maintained. The deep anesthesia was maintained with isoflurane (0.5% to 1.5%) mixed with oxygen during the neuronal recording. The neurons in the Vc/C1-C2, whose location was histologically confirmed after the experiment, were recorded using enamel-coated tungsten microelectrodes (impedance = 10–12 MΩ, 1,000 Hz; FHC, Bowdoin, ME, USA). Based on their responses to mechanical stimulation of the whisker pad skin, the recorded neuron was characterized as a nociceptive-specific neuron or a wide dynamic range neuron as previously described [50]. Background activity was recorded for 10 s prior to the mechanical stimulation of the whisker pad skin. Then, graded mechanical stimuli with von Frey filaments (1, 6, 15, 26, and 60 g) and a nylon hair brush were applied to the neuronal mechanoreceptive field (mRF) of the whisker pad skin for 5 s at 60 s intervals. High-intensity (pinch) stimulation produced by a small arterial clip was also applied to the neuronal mRF of the whisker pad skin for 5 s. The mean firing frequency (spikes/s) was calculated during mechanical stimulation. We defined neuronal activity with a mean firing rate twice the standard deviation of the mean background (BG) activity as neuronal spikes induced by mechanical stimulation. 

### 4.8. Statistical Analysis

All MHWT data are shown as median values in box-and-whisker plots. The box bottom and top indicate lower and upper quartiles, respectively. The lower and upper whiskers represent the minimum and maximum values, respectively. The one-way analysis of variance (ANOVA) followed by Tukey’s multiple comparison test was performed to compare the MHWTs following IONI with or without tranilast administration or following sham treatment. Other data are shown as the mean ± standard error (SEM). Student’s *t*-tests were performed to analyze the changes in IGF-1 expression, the number of FG-labeled TRPV2-IR and TRPV4-IR cells in each cell size, and the MHWTs following IONI with or without HC067047 administration. The one-way ANOVA followed by Tukey’s multiple comparison test was also performed for the immunohistochemical analysis and the analysis of changes in BG, brush-induced, and pinch-induced spike frequency. The two-way ANOVA with repeated measures followed by Tukey’s multiple comparison test was performed for the analysis of spike frequency induced by mechanical stimulation with von Frey filaments. A *p*-value of less than 0.05 was considered statistically significant.

## Figures and Tables

**Figure 1 ijms-20-06360-f001:**
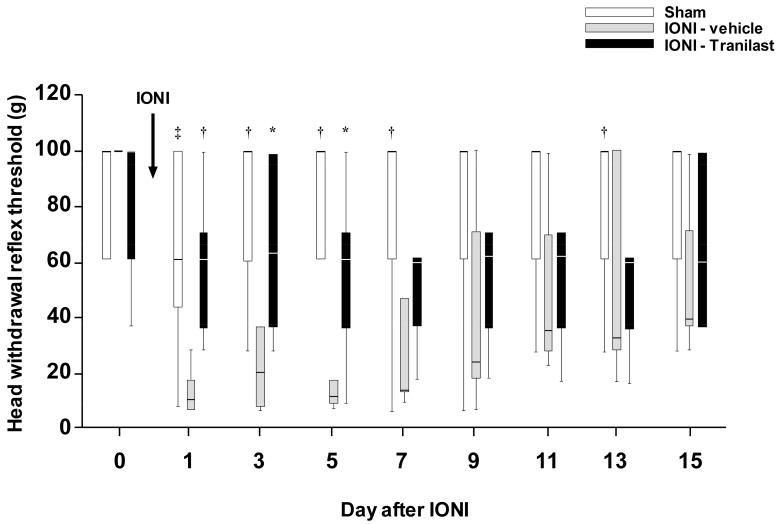
Changes in mechanical head withdrawal threshold of the whisker pad skin following sham treatment or IONI. Sham, sham treatment with vehicle administration; IONI-vehicle, IONI with vehicle administration; IONI-Tranilast: IONI with tranilast administration (*n* = 5 in each group). The box bottom and top indicate the lower and upper quartiles, respectively. The lower and upper whiskers represent the minimum and maximum values, respectively. * *p* < 0.05, † *p* < 0.01, ‡ *p* < 0.001 (vs IONI-vehicle group). IONI, infraorbital nerve injury.

**Figure 2 ijms-20-06360-f002:**
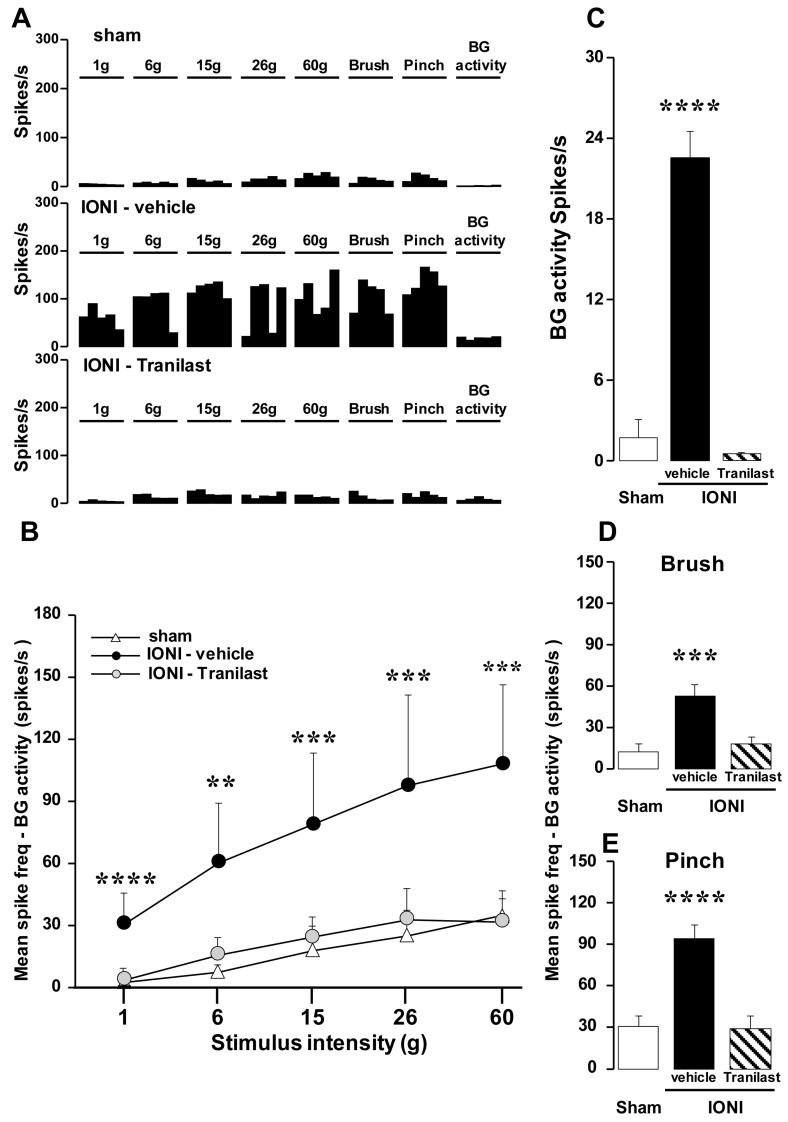
Activities of single WDR neurons in the upper cervical spinal cord on day 3. Peristimulus time histograms of WDR neurons in response to mechanical stimulation of the whisker pad skin and their BG activity on day 3 following sham treatment or IONI with or without tranilast administration (**A**). Mean spike frequencies of WDR neurons in response to graded mechanical stimulation of the whisker pad skin (**B**), no stimulation (**C**), and brush (**D**) or pinch (**E**) stimulation on day 3 following sham treatment and or IONI with vehicle or tranilast administration. ** *p* < 0.01, *** *p* < 0.001, **** *p* < 0.0001 (vs sham group). WDR, wide dynamic range; BG, background; IONI, infraorbital nerve injury.

**Figure 3 ijms-20-06360-f003:**
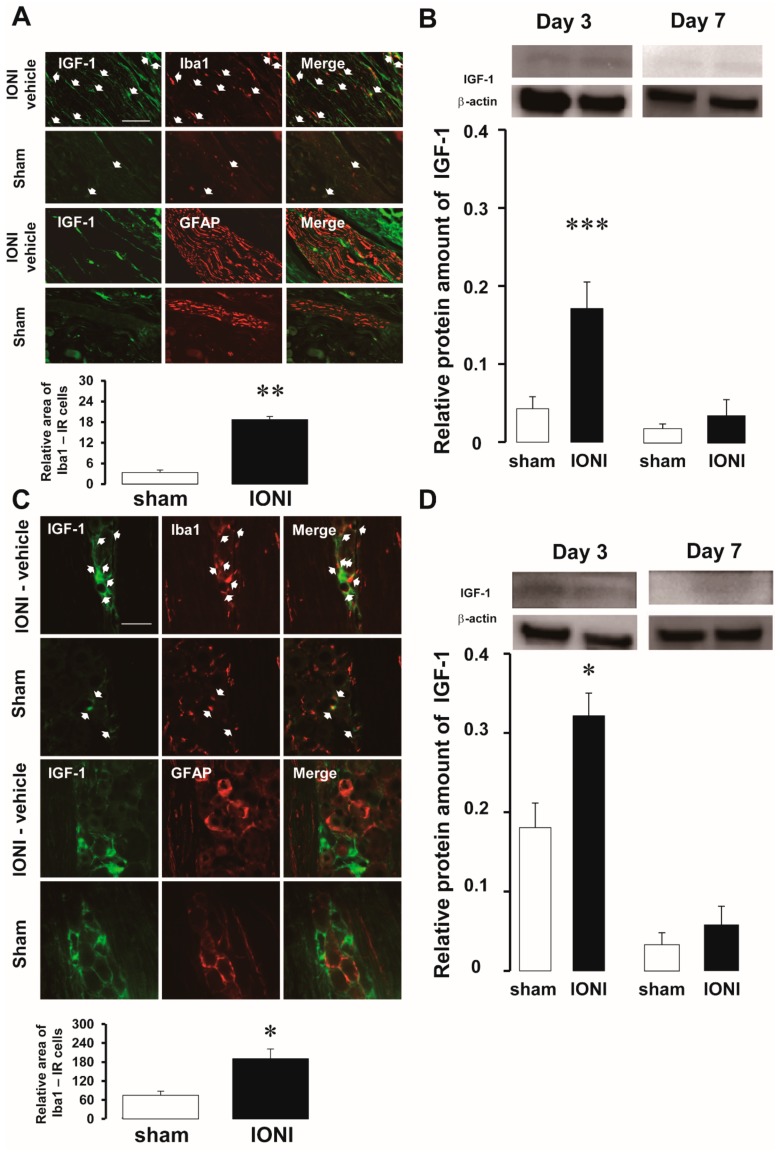
Peripheral IGF-1 expression following sham treatment or IONI. IGF-1 expression in Iba1- or GFAP-IR cells and the mean relative density of Iba1-IR cells in the injured ION on day 3 following IONI and sham treatment (**A**). Arrows indicate IGF-1- and Iba1-IR cells. Scale bar: 100 µm. ** *p* < 0.01 (vs sham treatment group). (*n* = 3 in each group). The amount of IGF-1 protein in the injured ION on day 3 and day 7 following sham treatment or IONI (**B**). The loading control was β-actin. *** *p* < 0.01 (vs sham treatment group; day 3, *n* = 9 in each group; day 7, *n* = 6 in each group). IGF-1 expression in Iba1-IR or GFAP-IR cells and the mean relative density of Iba1-IR cells in the TG on day 3 following IONI and sham treatment (**C**). Arrows indicate IGF-1- and Iba1-IR cells. Scale bar: 50 µm. * *p* < 0.05 (vs sham treatment group; *n* = 3 in each group). The amount of IGF-1 protein in the TG on day 3 and day 7 following sham treatment or IONI (**D**). The loading control was β-actin. * *p* < 0.05 (vs sham treatment group; day 3, *n* = 6 in sham group, *n* = 5 in IONI group; day 7, *n* = 6 in each group). IGF-1, insulin-like growth factor-1; IONI, infraorbital nerve injury; GFAP, glial fibrillary acidic protein; IR, immunoreactive; TG, trigeminal ganglion.

**Figure 4 ijms-20-06360-f004:**
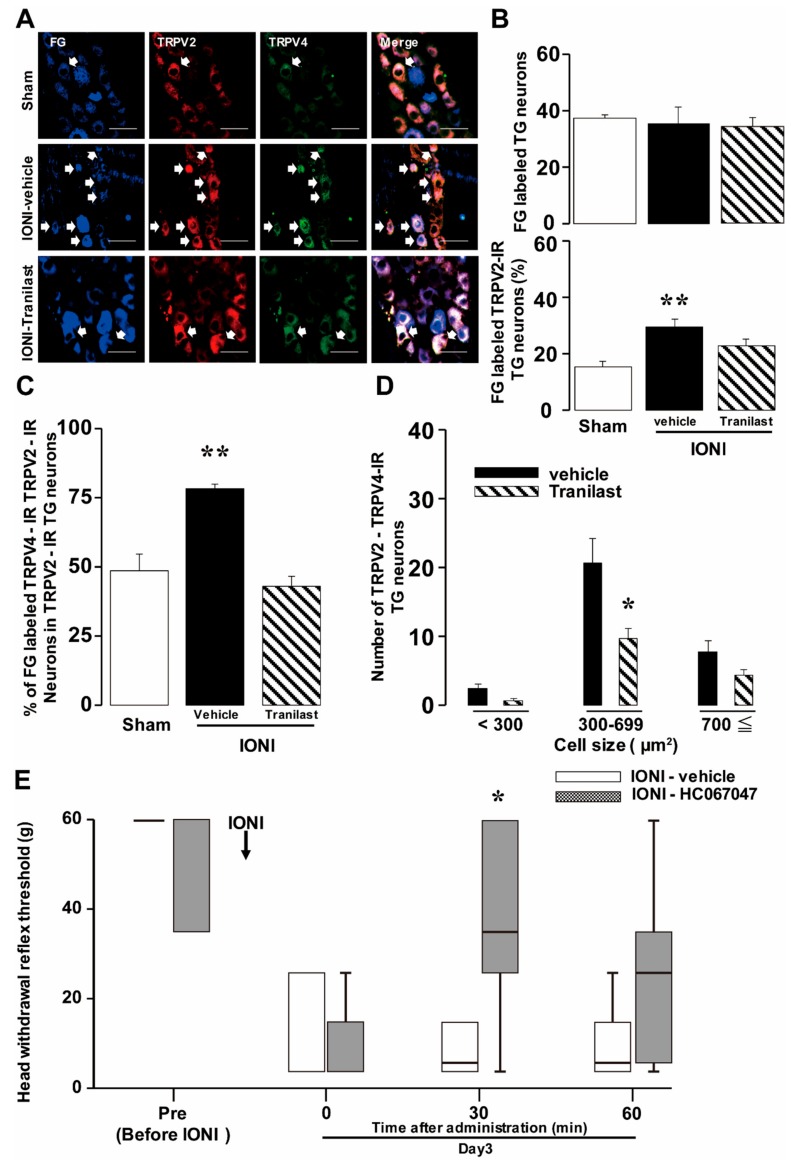
Changes in TRPV2 and TRPV4 expression in TG neurons and mechanical sensitivity due to TRPV4 antagonism on day 3 following IONI. TRPV2- and TRPV4-IR TG neurons innervating the whisker pad skin on day 3 following sham treatment or IONI with vehicle or tranilast administration (**A**). Arrows indicate FG-labeled TRPV2- and TRPV4-IR neurons. Scale bar: 50 µm. Mean percentages of FG-labeled TG neurons (upper) and FG-labeled TRPV2-IR TG neurons (lower) (**B**), mean percentages of FG labeled TRPV2- and TRPV4-IR TG neurons in FG-labeled TRPV2-IR TG neurons (**C**), and size-frequency histogram illustrating the distribution of TRPV2- and TRPV4-IR TG neurons (**D**) on day 3 following sham treatment or IONI with vehicle or tranilast administration. (*n* = 7 in each group). * *p* < 0.05, ** *p* < 0.01 (vs sham treatment group). (**E**) MHWTs of the whisker pad skin before and on day 3 following IONI with vehicle or HC-067047 administration to the whisker pad skin. The box bottom and top indicate lower and upper quartiles, respectively. The lower and upper whiskers represent the minimum and maximum values, respectively. (*n* = 5 in each group). * *p* < 0.05 (vs IONI-vehicle group). TRPV, transient receptor potential vanilloid subfamily; TG, trigeminal ganglion; IONI, infraorbital nerve injury; IR, immunoreactive; FG, FluoroGold; MHWT, mechanical head withdrawal threshold.

**Figure 5 ijms-20-06360-f005:**
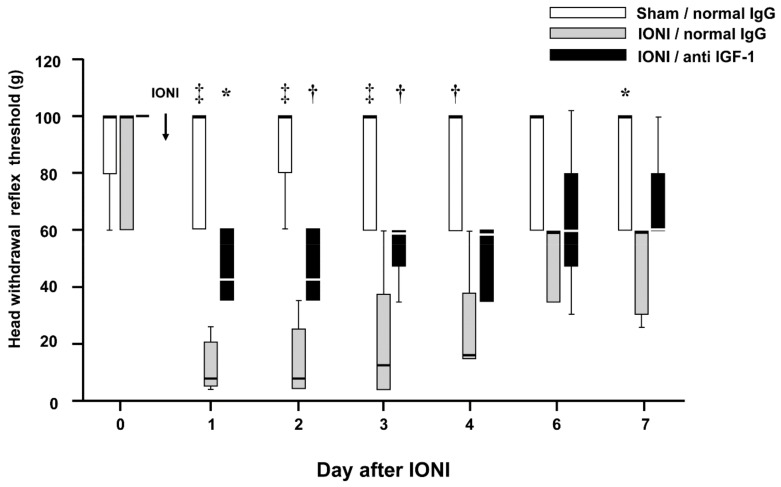
Changes in mechanical head withdrawal threshold of the whisker pad skin following sham treatment or IONI. Sham-normal IgG, sham treatment with normal IgG administration; IONI-normal IgG, IONI with normal IgG administration; IONI-anti IGF-1, IONI with IGF-1 neutralizing antibody administration (*n* = 5 in each group). The boxes bottom and top indicate the lower and upper quartiles, respectively. The lower and upper whiskers represent the minimum and maximum values, respectively. * *p* < 0.05, † *p* < 0.01, ‡ *p* < 0.001 (vs IONI-normal IgG group). IONI, infraorbital nerve injury.

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
