# Peer review of "Increase in IGF-1 Expression in the Injured Infraorbital Nerve and Possible Implications for Orofacial Neuropathic Pain"

_ijms, 2019, doi:10.3390/ijms20246360_

Round 1

Reviewer 1 Report

The present manuscript by Sugawara et al deals with potential causes for orofacial neuropathic pain following infraorbital nerve injury (IONI) in rats. Specifically, the authors examined the effects of transient receptor potential vanilloid (TRPV2) antagonism on orofacial mechanical hypersensitivity following IONI and quantified insulin-like growth factor-1 (IGF-1) expression in the injured ION and trigeminal ganglion (TG). In addition, they evaluated the effect of TRPV2 antagonism on the expression of the TRP vanilloid subfamily type 4 (TRPV4) channel, which is involved in nociceptive mechanical sensitivity to TG neurons innervating the orofacial region. The authors conclude that IGF-1 released by macrophages accumulate in the injured ION and binds to TRPV2, leading to an increase in TRPV4 expression in TG neurons. The net result is mechanical allodynia (central pain sensitization) of the whisker pad skin.

The paper is flawed in a number of fundamental respects, as detailed below.

Lines 50-52: the authors state that "IGF-1 is released by macrophages and also known as a potential ligand for the transient receptor potential vanilloid subfamily type 2 (TRPV2) channel ...". This statement is entirely incorrect. IGF-1 serves as a ligand for the IGF-1 receptor (IGF1R), AND NOT TRPV2. Activation (phosphorylation) of IGF1R by IGF-1 might eventually lead to activation of TRPV2, but that would be a secondary event.

Lines 181-182: the authors state that the amount of IGF-1 released by macrophages infiltrating the ION and TG was significantly increased on day 3 following IONI. Based on this single observation they conclude that IGF-1 is the mediator of the neuropathic pain. This conclusion is definitely wrong. Probably hundreds of different cytokines and growth factors (out of which they measured only IGF-1) are released in the injured area. One can not conclude this far-reaching conclusion from this immunohistochemistry experiment. Additional experiments are required to proof the role of IGF-1, including IGF1R silencing, IGF1 overexpression, etc etc

The title of the paper is incorrect. As described above, the authors fall short of demonstrating that IGF-1 contributes to orofacial neuropathic pain. Authors only show an increase in IGF-1 expression, which may be or may be not linked to neuropathic pain.

Figure 3D: the authors claim that Western blots show an increase in IGF-1 following IONI. However, this effect is due to a reduction in beta-actin levels (supposed to serve only as a loading marker) and not to a true increase in IGF-1.

Lines 224-228: the authors state that "IGF-1 released by macrophages in the injured ION and the corresponding TG contributes to increased expression and activation of TRPV4 in TG neurons ...". As detailed above, this conclusion is incorrect.

Author Response

Reviewer1

The present manuscript by Sugawara et al deals with potential causes for orofacial neuropathic pain following infraorbital nerve injury (IONI) in rats. Specifically, the authors examined the effects of transient receptor potential vanilloid (TRPV2) antagonism on orofacial mechanical hypersensitivity following IONI and quantified insulin-like growth factor-1 (IGF-1) expression in the injured ION and trigeminal ganglion (TG). In addition, they evaluated the effect of TRPV2 antagonism on the expression of the TRP vanilloid subfamily type 4 (TRPV4) channel, which is involved in nociceptive mechanical sensitivity to TG neurons innervating the orofacial region. The authors conclude that IGF-1 released by macrophages accumulate in the injured ION and binds to TRPV2, leading to an increase in TRPV4 expression in TG neurons. The net result is mechanical allodynia (central pain sensitization) of the whisker pad skin.

The paper is flawed in a number of fundamental respects, as detailed below.

Line 50-52: the authors state that “IGF-1 is released by macrophages and also known as potential ligand for the transient receptor potential vanilloid subfamily type 2 (TRPV2) channel …” This statement is entirely incorrect. IGF-1 serves as ligand for the IGF-1 receptor (IGF-1R), and not TRPV2. Activation (phosphorylation) of IGF-1R by IGF-1 might eventually lead to activation of TRPV2, but that would be a secondary event.

# We have modified the sentence as follows;

L52-55 “IGF-1 receptor (IGF-1R) activation by IGF-1 released from infiltrating macrophages induces translocation of transient receptor potential (TRP) vanilloid subfamily type 2 (TRPV2) channel to the cell membrane, resulting in noxious heat or mechanical hypersensitivity.”

Line 181-182: the authors state that the amount of IGF-1 released by macrophages infiltrating the ION and TG was significantly increased on day 3 following IONI. Based on this single observation they concluded that IGF-1 is the mediator of neuropathic pain. This conclusion is definitely wrong. Probably hundreds of different cytokines and growth factors (out of which they measured only IGF-1) are released in the injured area. One cannot conclude this far – reaching conclusion from this. Immumohistochenistry experiment, additional experiments are required to proof the role of IGF-1, including IGF-1R signaling IGF-1 overexpression etc,etc,

# To proof the role of IGF-1 signaling in the mechanical hypersensitivity following ION injury, we have performed an additional experiment and added Figure 5. Also, we have added and modified the sentences as follows;

L170-174 “2.5. Effects of IGF-1 neutralization on the Decreased MHWT Following IONI

To assess the effect of IGF-1 on the decreased MHWT following IONI, changes in the MHWTs after IONI by IGF-1 neutralization were examined. The decreased MHWTs were significantly recovered by daily IGF-1 neutralizing antibody administration (Day 1, sham - normal IgG: 76.00 ± 8.76 g, IONI - normal IgG: 11.40 ± 3.36 g, IONI - IGF-1 neutralizing antibody: 45.0 ± 5.48 g) (Fig. 5).”

L200-206 “Furthermore, IGF-1 neutralization partially depressed IONI-induced mechanical hypersensitivity. Taken together, our findings suggest that the infiltrating macrophages activated via CCL2-CCR2 signaling release IGF-1 following IONI, resulting in increased IGF-1 levels in the injured ION site and the ipsilateral TG. Increased IGF-1 may play an important role in IONI-induced mechanical hypersensitivity. However, it is expected that hundreds of different cytokines and growth factors which are released in the injured ION and the corresponding TG, further analysis is necessary.”

L303-310 “Furthermore, to evaluate that IGF-1 signaling in IONI-induced mechanical hypersensitivity of whisker pad skin, 10µl of rabbit IGF-1 neutralizing antibody (2.0 µg/ml, ab9572; Abcam, Cambridge, UK) or normal rabbit IgG (2.0 µg/ml, #148-09551; Wako, Tokyo, Japan) dissolved in 0.01 M PBS was subcutaneously administrated into whisker pad skin under inhalation anesthesia using 1.5% isoflurane once a day until day 7 following IONI using a 27-gauge needle. The MHWT was measured before and daily for 7 days following IONI or sham treatment under the same conditions as described above. The administered rabbit IGF-1 neutralizing antibody dose was based on previous reports [45, 46].”

Figure legend of Figure 5 “Figure 5. Changes in mechanical head withdrawal threshold of the whisker pad skin following sham treatment or IONI. Sham-normal IgG: sham treatment with normal IgG administration, IONI-normal IgG: IONI with normal IgG administration, IONI-anti IGF-1: IONI with IGF-1 neutralizing antibody administration (n = 5 in each group). The box bottom and top indicate the lower and upper quartiles, respectively. The lower and upper whiskers represent the minimum and maximum values, respectively. *p < 0.05, †p < 0.01, ‡p < 0.001 (vs IONI-normal IgG group). IONI, infraorbital nerve injury.”

The title of this paper is incorrect. As described above, the authors fall short of demonstrating that IGF-1 contributes to orofacial neuropathic pain. Authors only shown an increased in IGF-1 expression, which may be or may be not linked to neuropathic pain.

# We have modified the title as follows;

L2-4 “Increase in IGF-1 expression in the injured infraorbital nerve and possible implications for orofacial neuropathic pain.”

Figure3D: The authors claim that Western blots show an increased in IGF-1 following IONI. However this effect is due to a reduction in beta-actin level (supposed to serve only as a loading marker ) and not to a true increase in IGF-1.

# We have replaced the Western blot image in Figure 3D.

Line224-228: the authors state that “IGF-1 released by macrophages in the injured ION and the corresponding TG contributes to increased expression and activation of TRPV4 in TG neurons…” As detailed above, this conclusion is incorrect.

# We have modified the sentence as follows;

L246-249 “these results suggest that IGF-1 released by macrophages in the injured ION and the corresponding TG anticipates TRPV4 activation in TG neurons innervating the whisker pad skin via IONI-induced TRPV2 signaling, which is a potential mechanism of mechanical allodynia of the whisker pad skin.”

Reviewer 2 Report

The paper by Sugawara and coworkers was aimed at studying the involvement of IGF-I produced by macrophages, that accumulates at the site of infraorbital nerve lesion, in the development of allodynia at the wisker pad skin via TRPV2 signaling.

The paper is quite interesting, but the evidence of the involvement of IGF-I has some weakness.

IGF-I levels seem too much variable, as shown in sham rats (values in sham 0.042 vs controlateral site, sham 0.18). Do you have some plausible explanations for this variability?

In addition, the determinations of IGF-I protein, demonstrating increased levels at the site of the lesion, were performed only three days after the nerve lesion. A time-course evaluation could be much more relevant to reinforce the hypothesis that IGF-I may contribute to the development of neuropathic pain after nerve lesion.

Author Response

Reviewer2

The paper by Sugawara and coworkers was aimed at studying the involvement of IGF-I produced by macrophages, that accumulates at the site of infraorbital nerve lesion, in the development of allodynia at the whisker pad skin via TRPV2 signaling.

The paper is quite interesting, but the evidence of the involvement of IGF-1 has some weakness.

# To proof the role of IGF-1 signaling in the mechanical hypersensitivity following ION injury, we have performed an additional experiment and added Figure 5. Also, we have added and modified the sentences as follows;

L170-174 “2.5. Effects of IGF-1 neutralization on the Decreased MHWT Following IONI

To assess the effect of IGF-1 on the decreased MHWT following IONI, changes in the MHWTs after IONI by IGF-1 neutralization were examined. The decreased MHWTs were significantly recovered by daily IGF-1 neutralizing antibody administration (Day 1, sham - normal IgG: 76.00 ± 8.76 g, IONI - normal IgG: 11.40 ± 3.36 g, IONI - IGF-1 neutralizing antibody: 45.0 ± 5.48 g) (Fig. 5).”

L200-206 “Furthermore, IGF-1 neutralization partially depressed IONI-induced mechanical hypersensitivity. Taken together, our findings suggest that the infiltrating macrophages activated via CCL2-CCR2 signaling release IGF-1 following IONI, resulting in increased IGF-1 levels in the injured ION site and the ipsilateral TG. The increased IGF-1 may play an important role in IONI-induced mechanical hypersensitivity. However, it is expected that hundreds of different cytokines and growth factors which are released in the injured ION and the corresponding TG, further analysis is required.”

L303-310 “Furthermore, to evaluate that IGF-1 signaling in IONI-induced mechanical hypersensitivity of whisker pad skin, 10µl of rabbit IGF-1 neutralizing antibody (2.0 µg/ml, ab9572; Abcam, Cambridge, UK) or normal rabbit IgG (2.0 µg/ml, #148-09551; Wako, Tokyo, Japan) dissolved in 0.01 M PBS was subcutaneously administrated into whisker pad skin under inhalation anesthesia using 1.5% isoflurane once a day until day 7 following IONI using a 27-gauge needle. The MHWT was measured before and daily for 7 days following IONI or sham treatment under the same conditions as described above. The administered rabbit IGF-1 neutralizing antibody dose was based on previous reports [45, 46].”

Figure legend of Figure 5 “Figure 5. Changes in mechanical head withdrawal threshold of the whisker pad skin following sham treatment or IONI. Sham-normal IgG: sham treatment with normal IgG administration, IONI-normal IgG: IONI with normal IgG administration, IONI-anti IGF-1: IONI with IGF-1 neutralizing antibody administration (n = 5 in each group). The box bottom and top indicate the lower and upper quartiles, respectively. The lower and upper whiskers represent the minimum and maximum values, respectively. *p < 0.05, †p < 0.01, ‡p < 0.001 (vs IONI-normal IgG group). IONI, infraorbital nerve injury.”

IGF-1 level seen too much variable, as shown in sham rats (value in sham 0.042 vs contralateral site, sham 0.18) Do you have plausible explanation for this variability?

We have compared IGF-1 level in the injured-ION or TG ipsilateral to IONI and IGF-1 level in the uninjured-ION or TG ipsilateral to the uninjured-ION. Therefore, we have not compared value in sham and contralateral site. The difference in IGF-1 level is thought to be due to the collected tissues (ION and TG). Indeed, immunohistochemical staining in Figure 3 shows that IGF-1 expression in the uninjured-ION (sham) is weaker than that of TG ipsilateral to the uninjured-ION (sham).

In addition, this determination of IGF-1 protein, demonstrating increased levels at the site of the lesion, were performed only three days after the nerve lesion. A time- course evaluation could be much more relevant to reinforce the hypothesis that IGF-1 may contribute to the development of neuropathic pain after nerve lesion.

# We have performed an additional experiment to examine the IGF-1 level in the injured-ION on day 7 after IONI. We have added and modified the sentences and the phrases as follows;

L116-118 “The relative amount of IGF-1 protein in the injured ION and TG ipsilateral to IONI recovered to the same level as sham treatment on day 7 following IONI.”

L352-353 “On days 3 and 7 after IONI or sham treatment, the rats were perfused with physiological saline under deep i.p. anesthesia with the above-described solution.”

Figure legend of Figure 3 “The amount of IGF-1 protein in the injured ION on day 3 and day 7 following sham treatment or IONI (B).”

“(day 3: n = 9 in each group; day 7: n = 6 in each group)”

“The amount of IGF-1 protein in the TG on day 3 and day 7 following sham treatment or IONI (D).”

“(day 3: n = 6 in sham group, n = 5 in IONI group; day 7: n=6 in each group)”

Round 2

Reviewer 1 Report

The authors have satisfactorily addressed reviewer's concerns.

Reviewer 2 Report

Authors provided some answers and added new experimental data to their original manuscript.

In this version, the paper could have more potential interest. It is not possible finding a definitive answer on the role of IGF-1 in the neuropathic pain, however, the paper could stimulate further researches and discussions on the topic, contributing, likely, to find more interesting data than that data present in this paper.
I suggest to publish the manuscript.